# Selective decontamination of the digestive tract in esophagectomy and the incidence of pneumonia and anastomotic leakage: A systematic review and meta-analysis

Sander Du X Oei[1][☯], Jasper Gerrit Jan Verbruggen[1][☯], Sanne Elisabeth Hoeks[1], Marcus Paulus Buise[2]*

1 Department of Anesthesiology, Erasmus University Medical Center, Rotterdam, The Netherlands,
2 Department of Anesthesiology, Maastricht University Medical Center, Maastricht, The Netherlands

☯ These authors contributed equally to this work.
* marc.buise@mumc.nl

## Abstract

### Background

Despite advances in surgery, esophagectomy remains a major operation in which pneumonia and anastomotic leakage are causes of morbidity. It is currently unknown whether selective decontamination of the digestive tract (SDD) affects the incidence of postoperative pneumonia and anastomotic leakage in patients undergoing esophagectomy. The aim of this systematic review and meta-analysis is to summarize current evidence regarding SDD in patients undergoing esophagectomy.

### Methods

We performed a comprehensive search in Medline, Web of Science, Embase, Cochrane Library and Google Scholar with articles included until August 2024. We included observational studies and clinical trials which were scored using the Cochrane Risk of Bias tool and The Risk Of Bias In Non-randomized Studies – of Interventions. A fixed effects model was used to pool results of the former studies.

### Results

A total of five studies were identified with a total of 924 patients. All studies were assessed as either having serious bias or a high risk of bias. SDD usage was associated with a significantly lower incidence of pneumonia (OR 0.41; 95% CI 0.29 to 0.58; $p < 0.00001$; $I^2 = 26\%$; $n = 924$) and anastomotic leakage (OR 0.48; 95% CI 0.30 to 0.74; $p = 0.001$; $I^2 = 0\%$; $n = 810$). Pooled analysis regarding mortality, duration of hospitalization and duration of Intensive Care Unit stay could not be performed due to heterogeneous data, 4 of 5 studies reported lower mortality rates in patients receiving SDD.

**Data availability statement:** All raw data required to replicate the study is available in the paper and Supporting Information.

**Funding:** The author(s) received no specific funding for this work.

**Competing interests:** We have no conflicts of interests. M. Buise is part of an international think tank regarding minimally invasive esophagectomy which is supported by Medtronic. This does not alter our adherence to PLOS ONE policies on sharing data and materials.

## Conclusion

Although the data indicates that using SDD in patients undergoing an esophagectomy was associated with a lower incidence of postoperative pneumonia and anastomotic leakage, the available studies were not of sufficient quality to make a recommendation, given their age and risk of bias. A high-quality randomized controlled trial using standardized outcome definitions is needed to substantiate claims about SDD use in esophagectomy.

## Introduction

The incidence of esophageal cancer is rising, globally 604 100 cases were diagnosed in 2020 making it the eight most diagnosed form of cancer [1]. Treatment modalities include chemotherapy, radiotherapy, immunotherapy and surgical resection [2]. Esophagectomy is considered to be standard of care for curative treatment [3,4].

Surgical esophagectomy techniques may differ based on tumor location, necessity of two-field or three-field lymphadenectomy and operator preference [5]. Techniques may include transhiatal esophagectomy, transthoracic esophagectomy or en bloc esophagectomy, possibly accompanied with two-field or three-field lymphadenectomy [6]. Minimally invasive esophagectomy (MIE) techniques are now the most common approach worldwide [7]. MIE has similar disease-free survival compared to open techniques and lower rates of postoperative complications [8,9].

Postoperative morbidity following esophagectomy is, although reduced, still high despite MIE techniques. Postoperative pulmonary complications (PPCs) occur in approximately 30% of open esophagectomies and 18% in MIE [10]. Anastomotic leakage occurs in 6–19.9% of esophagectomies regardless of chosen technique (robot-assisted minimally invasive esophagectomy or open esophagectomy) [10–12]. These complications are independent risk factors associated with an increased 30-day postoperative mortality [13].

The high incidence of PPCs is multifactorial, it is probably partly due to (micro) aspirations as a result of delayed gastric emptying in the early post-operative period [14,15]. Ischemia of the tip of the gastric conduit is one of the major contributing factors to anastomotic leakage after esophagectomy [16]. An additional factor that possibly contributes to anastomotic leakage is the presence of bacteria that produce endo- or exotoxins. Certain bacteria, such as *P. aeruginosa*, may interfere with the healing of the anastomosis and cause anastomotic leakage through downregulation of fibroblast growth, production of toxins, and infection, which lead to necrosis (also promoted via microcirculatory disturbances) [17]. Intestinal healing of the anastomosis and PPCs may be compromised due to certain specific bacterial families [18,19].

Selective decontamination of the digestive tract (SDD) involves the use of non-resorbable antibiotics, typically administered orally, to prevent pathogenic Gram-negative infections while preserving the normal anaerobic intestinal flora. These antibiotics are most commonly administered both orally and via a gastric tube, if present. The non-absorbable antibiotics, polymyxin E and tobramycin target a broad

spectrum of potentially harmful aerobic Gram-negative rods and *S. aureus*. Amphotericin B is often added to prevent the overgrowth of molds and yeasts [20]. SDD differs from standard perioperative antibiotic prophylaxis, which is typically administered intravenously at the time of induction and generally lasts for a shorter duration (e.g., a single-dose administration during surgery or 24–48 hours of postoperative coverage).

Selective decontamination of the digestive tract (SDD) has been associated with improved patient outcome in critically ill patients [20]. A systematic review of perioperative SDD in elective gastrointestinal surgery suggested that a combination of perioperative SDD and perioperative intravenous antibiotics reduces the rate of postoperative infection and anastomotic leakage compared with systemic antibiotic prophylaxis alone [21].

Because micro-aspirations may often occur after esophagectomy, treatment with SDD in this patient group is of particular interest [22,23].

It is currently unknown whether selective decontamination of the digestive tract affects incidence of postoperative pneumonia and anastomotic leakage in patients undergoing esophagectomy. The aim of this paper is to perform a systematic review and meta-analysis of current evidence comparing SDD versus regular care or placebo in patients undergoing esophagectomy.

## Methods

This systematic review was registered in the International Prospective Register of Systematic Reviews PROSPERO (CRD42022333140). Data reporting and review are consistent with the Preferred Reporting Items for Systematic Reviews and Meta-Analyses (PRISMA) statement (S1 File) [24].

### Literature search

A comprehensive search was performed in Medline, Web of Science, Embase, Cochrane Library and Google Scholar in cooperation with a qualified medical librarian, the search strategy is available in the S2 File. All results up to August 2024 were included without any language restrictions. Reference lists were also manually checked for further potentially relevant studies using the snowballing technique. All articles found were screened on title and abstract. Two authors (SO and JV) independently screened titles/abstracts and full text articles and discrepancies were resolved by a third author (MB). All studies that were not excluded in the screening stage were assessed in full text for eligibility by two authors independently (SO and JV). This process will be described in a flow chart according to the PRISMA statement [24].

### Selection criteria

Databases were searched for articles on esophagectomy, regardless of the surgical technique used or the year of publication. To obtain a comprehensive view of the literature that could be effectively combined in a meta-analysis, we included both observational and interventional studies in the search, while excluding case series and case reports. The studies had to report on postoperative pulmonary complications or anastomotic leaks to be included, which were the primary outcomes of interest for our review.

### Quality assessment

All studies were independently assessed for methodological quality by SO and JV, any discrepancies that arose were discussed between authors. Persisting discrepancies were resolved by consulting a third author (MB).

The risk of bias for randomized trials was assessed using the Cochrane Risk of Bias tool (RoB 2) which contains the following components; randomization process, deviations from the intended interventions, missing outcome data, measurement of the outcome, selection of reported results [25]. The Risk Of Bias In Non-randomized Studies – of Interventions (ROBINS-I) tool was used to assess bias in non-randomized studies, which assessed the following components;

bias due to confounding, bias due to participant selection, bias in classification of interventions, bias due to deviations from intended interventions, bias due to missing outcome data, bias in measurement of outcomes, bias in selection of the reported results [26].

Studies were classified as either low, moderate or serious for each component of the ROBINS-I and RoB 2 tools and an overall bias classification was also reported.

### Outcome parameters and data extraction

The primary outcomes of interest were postoperative pulmonary complications and anastomotic leakage. Secondary outcome parameters were ICU length of stay, hospital length of stay and mortality. Authors of included studies were contacted when outcome data was not reported.

Additional information extracted included study design, number of included patients, SDD components used, SDD regimen, study period, age, gender distribution, BMI and ASA classification.

### Statistical analysis

Statistical analyses were performed using the Cochrane Software Review Manager (version 5.4, The Cochrane Collaboration, Copenhagen, Denmark).

Meta-analysis was performed when data on an outcome parameter was reported in at least two studies in a way that was compatible with meta-analysis. A subgroup analysis with solely the randomized patients was executed. Dichotomous data was analyzed using the Mantel-Haenszel method, presented as odds ratio (OR) along with 95% confidence intervals (CI). Assessment for publication bias was qualitative through visual inspection for funnel plot asymmetry, as can be seen in the S3 File. Depending on statistical and methodological heterogeneity a fixed-effect or random effects model was used. Statistical heterogeneity was assessed with $I^2$. In the absence of substantial statistical heterogeneity ($I^2 \leq 50\%$) a fixed-effect model was used [27]. In case of substantial heterogeneity ($I^2 > 50\%$), a random effects model was used. p values $\leq 0.05$ were considered statistically significant.

## Results

### Study characteristics

In summary, five articles meeting our search criteria were initially identified from five databases [12,28–31].

Five articles published between 1990 and 2021 involving 924 participants were included (Fig 1). The characteristics of the eligible articles are summarized in Table 1.

Preoperative antibiotic prophylaxis was administered in all groups in all studies, independent of assignment to SDD. Antibiotic regimens differed, one study used amoxicillin/clavulanic acid [29], all others administered a cephalosporin, of which 3 also administered metronidazole [12,28,31].

The SDD regimen differed between trials. Tetteroo and Riedl used a combination of polymyxin, tobramycin and amphotericin [28,30]. Farran and colleagues used a combination of erythromycin, gentamicin and nystatin [29]. Janssen and colleagues used amphotericin, polymyxin and tobramycin [12]. Näf and colleagues used a combination of polymyxin, tobramycin, nystatin and vancomycin thereby covering methicillin-resistant *S. aureus* (MRSA) [31].

### Quality assessment

Based on the ROBINS-I [26], both non-randomized studies were classified as being at serious risk of bias, see Table 2. Both studies were classified as having a serious risk of bias due to a potential for confounding.

Based on the RoB 2 [25], all studies were classified as being at high risk of bias, see Table 3. All randomized studies had a low chance of bias due to missing data, but risk of bias in all other domains (randomization, deviations from the

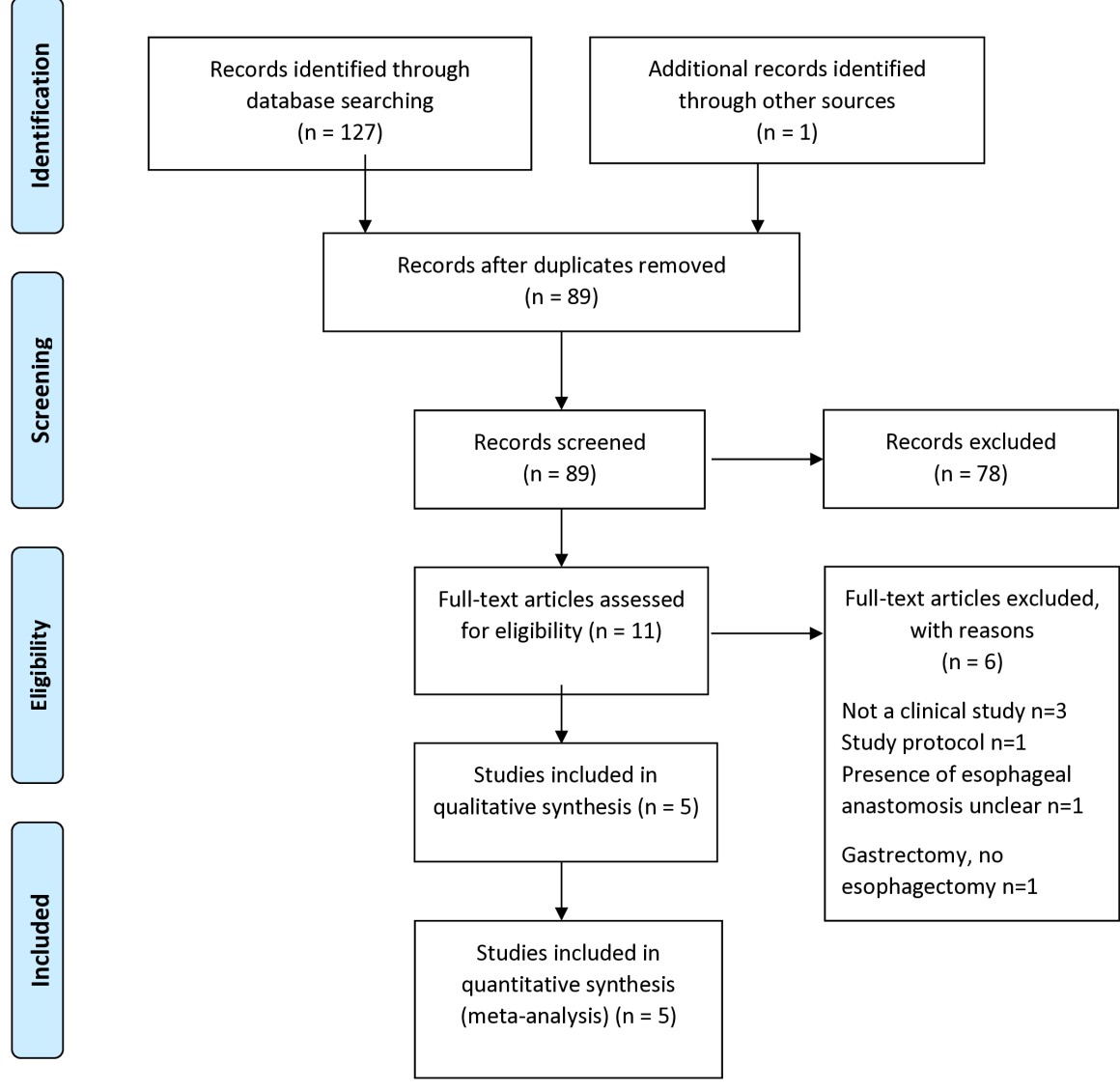

**Fig 1. PRISMA flow diagram.**

intended interventions, measurement of outcome and selection of reported results) ranged from 'some concern' to 'high risk'. The detailed bias assessment can be found in the S4 File.

## Primary outcomes

**Impact of SDD on pneumonia and anastomotic leakage.** Five studies reported on the incidence of postoperative pulmonary complications. Tests of heterogeneity showed little heterogeneity between the studies. Visual inspection of the funnel plot showed no asymmetry (S3 File). Results are displayed in Fig 2. Overall pooled analysis indicated a lower incidence of postoperative pneumonia (OR 0.41; 95% CI 0.29 to 0.58; p < 0.00001; $I^2 = 26\%$; n = 924, 5 studies) for patients treated with SDD. In the subgroup of randomized studies, a comparable lower risk of pneumonia was seen in patients who received SDD (OR 0.39; 95% CI 0.16 to 0.95; p = 0.04; $I^2 = 7\%$; n = 184, 3 studies).

**Table 1. Characteristics of included studies.**

| Study | Study design | Number of patients (SDD vs control) | Operation technique | Age[a] | ASA | Tumor stage SDD | Control | Male sex | BMI (SD) | Antibiotics used | Regimen | Outcome parameters |
|---|---|---|---|---|---|---|---|---|---|---|---|---|
| Tetteroo 1990 | RCT | 114 (56 vs 58) | Laparotomy, thoracotomy, cervical incision | 60 (39-78) | n.r. | n.r. | n.r. | 69% | n.r. | Polymyxin E 200mg, tobramycin 80mg, amphotericin B 500mg and systemic cefotaxime 1000mg. | Admission until 10th postoperative day. Cefotaxime iv 3 days before surgery until 1 day after. 4 times daily. | Postoperative infections (clinical and microbiological signs of infection). |
| Riedl 2001 | Partially randomized patient preference trial | 70 (25 vs 45) | Two-field | 56 (9.4) | n.r. | n.r. | n.r. | 92% | n.r. | Polymyxin B 100mg, tobramycin 80mg and amphotericin B 200mg. | 4-7 days before surgery until at least 1wk after. 4 times daily. | Infection rate (Robert Koch Institute's definition), ICU stay, mortality. |
| Farran 2008 | RCT | 42 (18 vs 24) | Open Ivor-Lewis/open Akiyama | 60 (31-87) | n.r. | n.r. | n.r. | 95.2% | n.r. | Erythromycin 500mg, gentamicin 80mg and nystatin 100mg. | 12h before surgery until 5th day after. 4 times daily. | Postoperative pneumonia (clinical, radiological or microbiological signs), AL (ethylene blue or oral contrast), mortality. |
| Näf 2010 | Prospective cohort with retrospective control groups | 202 (134 vs 68) | Total gastrectomy, transhiatal extended gastrectomy, Merendino procedure | 63 (12.0) | I – 4% II – 63% ≥ III – 33% | n.r. | n.r. | 42%[b] | 24.6±4.2 | Polymyxin 100mg, tobramycin 80mg, vancomycin 125mg and nystatin 500mg. | Day before surgery until 7th postoperative day. 4 times daily. | Pulmonary infection (clinical, radiological or microbiological signs), AL (oral contrast imaging), mortality, LOS. |
| Janssen 2021 | Retrospective cohort | 496 (179 vs 317) | Ivor-Lewis TMIE | 65 (8.4) | I – 12% II – 62% ≥ III – 26% | I 11.4% II 40.9% III 47.7% | I 32.8% II 46.8% III 20.4% | 84% | 26.0±4.1 | Amphotericin B 500mg, polymyxin E 100mg and tobramycin 80mg. | 3 days before surgery until 3 days after. 4 times daily. | Pneumonia (according to rUPS), AL (irrespective of method of identification), mortality, LOS. |

[a] range or SD as reported in study

[b] sex reported in prospective cohort only

SDD, selective decontamination of the digestive tract; SD, standard deviation; ASA, American Society of Anesthesiology physical status; BMI, body mass index; RCT, randomized controlled trial; n.r., not reported; AL, anastomotic leakage; ICU, Intensive Care Unit; LOS, Length of stay; TMIE, totally minimally invasive esophagectomy; rUPS, revised Uniform Pneumonia Sc

**Table 2. The Risk Of Bias In Non-randomized Studies – of Interventions (ROBINS-I).**

| Study | Confounding | Selection | Classification | Deviations | Missing data | Measurement of outcomes | Selection of reported result | Overall |
|-------|-------------|-----------|----------------|------------|--------------|--------------------------|------------------------------|---------|
| Näf | Serious | Low | Low | NI | Low | Low | Low | Serious |
| Janssen | Serious | Low | Low | NI | Low | Low | Moderate | Serious |

**Table 3. Cochrane Risk of Bias tool (RoB 2).**

| Study | Randomization | Deviations | Missing data | Measurement of the outcome | Selection of the reported result | Overall |
|-------|---------------|------------|--------------|----------------------------|----------------------------------|---------|
| Tetteroo | Some concerns | High risk | Low risk | High risk | Some concerns | High risk |
| Riedl | High risk | High risk | Low risk | Some concerns | Some concerns | High risk |
| Farran | Some concerns | Some concerns | Low risk | High risk | Some concerns | High risk |

A total of four studies [12,29–31] with 810 patients reported anastomotic leakage. Tests of heterogeneity showed little heterogeneity between the studies. Visual inspection of the funnel plot showed no asymmetry (S3 File). Overall pooled analysis indicated a lower incidence of anastomotic leak when SDD was used perioperatively (OR 0.48; 95% CI 0.30 to 0.75; p = 0.001; I² = 0%; n = 810, 4 studies). In the subgroup of randomized studies, the risk of anastomotic leakage in randomized trials was not significantly different (OR 0.38; 95% CI 0.08 to 1.79; p = 0.22; I² = 0%; n = 70, 2 studies).

### Secondary outcomes

**Impact of SDD on mortality and length of stay.** Mortality rates were reported in all included studies, pooled analysis could not be performed due to varying reported time periods. Results are outlined in Fig 3 and Table 4. All studies except Tetteroo showed lower mortality rates in patients receiving SDD. The only statistically significant mortality difference was seen in the study performed by Näf et al. (30-day mortality 1.5% vs 17.6% favoring the SDD group).

Time period not specified in Tetteroo, Riedl and Farran. 30-day mortality reported in Näf and Janssen

Hospital length of stay was not pooled due to differences in reporting means and medians per included study. Hospital length of stay was generally longer in groups not receiving SDD except for the study performed by Janssen et al (13 vs 11) and in the randomized subgroup of Riedl (34 vs 28.5 days). ICU length of stay was slightly longer for patients in the SDD subgroups. Within each group, there was considerable variation in ICU LOS and hospital LOS.

## Discussion

Based on the results of previous studies, we comprehensively analyzed the effects of peri-operative SDD use on the incidence of PPCs and AL after esophagectomy. Our meta-analysis included 5 studies involving 924 patients in total. The strengths of our systematic review and meta-analysis include an extensive search across multiple databases, which yielded a thorough and reliable summary of the current state of knowledge in this area. Our review has several limitations; in the following paragraphs, we outline the factors that should be considered when interpreting our findings.

In this systematic review the use of peri-operative SDD was associated with a lower incidence of postoperative pneumonia and anastomotic leak after esophagectomy. However, analysis of the included studies in our meta-analysis suggests that all included studies are susceptible to at least some form of bias, which could potentially limit the reliability of the findings. A recent systematic review investigating the use of standardized clinical pathways on esophagectomy outcomes showed incidences of complications that correspond reasonably well with our data [32]. It is important to note

### A. Postoperative pneumonia

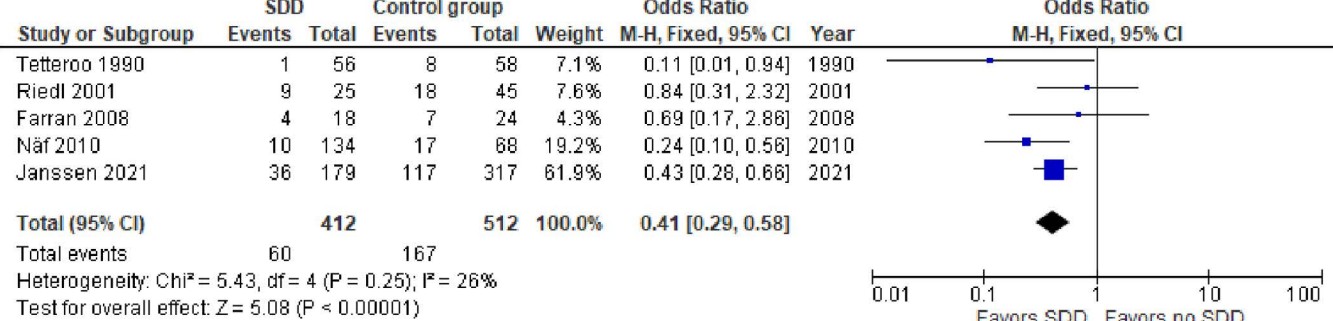

### B. Anastomotic leakage

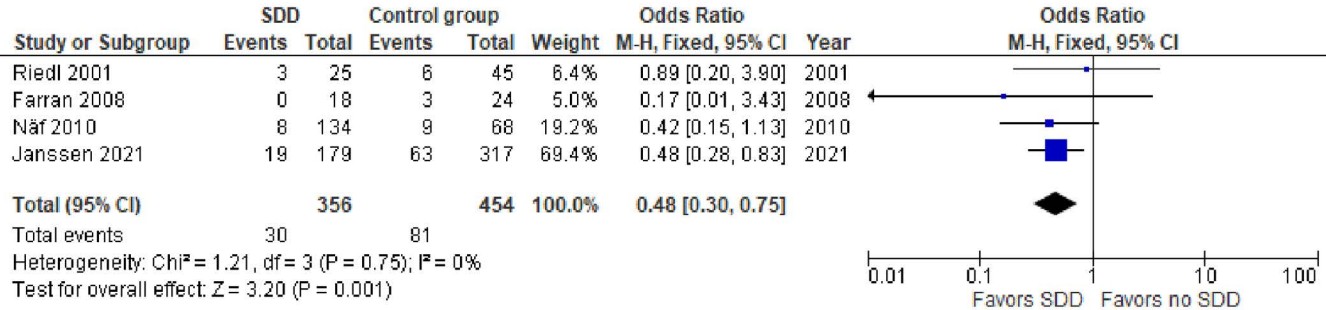

### A. Postoperative pneumonia, randomized trials only

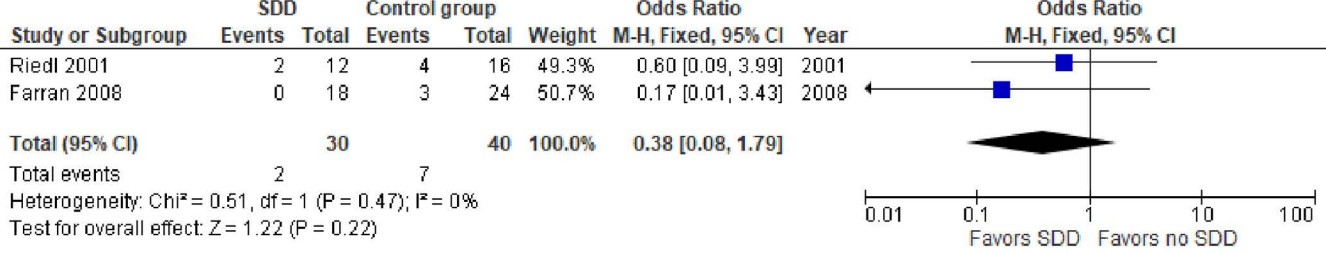

### B. Anastomotic leakage, randomized trials only

**Fig 2. Forest plots comparing the impact of SDD on pneumonia and anastomotic leakage.**

that some of the articles used in our review are dated. Although patient care has changed over the years, including the introduction of new minimally invasive surgical techniques, sparing of the pulmonary vagal branches, Enhanced Recovery After Surgery (ERAS) and prehabilitation the mechanisms of postoperative pneumonia and AL (micro-aspirations and tissue perfusion) remain unchanged [10,33–38].

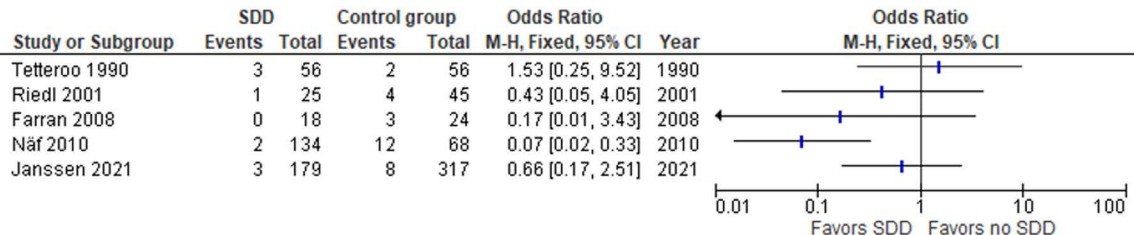

**Fig 3. Forest plot comparing the impact of SDD on mortality.**

**Table 4. Impact of SDD on mortality and LOS.**

| Reference | Mortality | | ICU LOS | | Hospital LOS | |
|---|---|---|---|---|---|---|
| | SDD | No SDD | SDD | No SDD | SDD | No SDD |
| Tetteroo (n = 114) | 5.4%[a] | 3.6%[a] | 5.5 (1–99) | 4.9 (0–35) | 25.6 (14–103) | 27.7 (11–91) |
| Riedl (n = 70) | 4%[a] | 8.9%[a] | 7 (1–27) | 6 (1–14) | 27 (17–74) | 31 (15–31) |
| Riedl randomized patients (n = 28) | 0%[a] | 6.3%[a] | 8 (3–15) | 5.5 (3–74) | 34 (18–74) | 28.5 (15–113) |
| Farran (n = 42) | 0%[a] | 12.5%[a] | n.r. | n.r. | 22 | |
| Näf (n = 202) | 1.5%[b,c] | 17.6%[b] | n.r. | n.r. | 20.8 (12.1) | 25.2 (18.0) |
| Janssen (n = 496) | 1.7%[b] | 2.5%[b] | 4 (2–7) | 1 (1–2) | 13 (10–20) | 11 (8–19) |

[a] unspecified time period, [b] 30-day mortality, [c] data from retrospective and prospective SDD cohorts combined.

SDD; Selective decontamination of the digestive tract, LOS; length of stay, ICU; intensive care unit, n.r.; not reported.

Incidence of postoperative pneumonia and AL remain high in present years despite minimally invasive surgical techniques [32]. ICU length of stay, duration of mechanical ventilation, and overall hospital length of stay have decreased substantially, but incidences of pneumonia and anastomotic leak were comparable amongst all the studies included in our analysis. The high incidence of postoperative pulmonary complications does therefore not seem attributable to ventilator-associated pneumonia. With improvements in techniques to the current standard, triggering factors for pulmonary complications may have decreased. Therefore, aspiration may now play a relatively larger role as a cause of pulmonary complications compared to the period of some studies included in our review. Swallowing dysfunction and tracheobronchial aspiration occur in a significant number of patients after esophagectomy. The mechanism is thought to be related to swallowing dysfunctions, possibly exacerbated by recurrent laryngeal nerve palsy [23,39]. Silent aspiration occurs in 44.7% of patients who had been confirmed as having aspiration with videofluoroscopic swallowing studies [40]. Furthermore, the anatomic location of the gastric conduit may play an etiological role in aspiration. Bulging of the gastric conduit into the right chest likely delays gastric emptying leading to a higher rate of lesions on computed tomography [41]. The management of delayed gastric emptying remains a topic of ongoing debate [42]. Routine upper gastrointestinal contrast studies are not currently recommended in the ERAS guidelines, although their use in identifying patients with delayed gastric conduit emptying shows promise [36,43].

Although the results of all studies indicate a benefit for SDD, it should be noted that our results are most influenced by the study performed by Janssen et al, the most recent study with the largest number of patients. It is worth following up this retrospective study with a double blind RCT to validate its findings. The PERSuaDER-trial, although not blinded, might provide more concrete evidence regarding perioperative SDD for esophagectomy [44].

Mortality rates in the included studies were inconclusive. While most studies found favorable mortality rates in patients receiving SDD, Tetteroo found the opposite and in Janssen there was a trend towards a higher 90-day and 1-year mortality in the SDD group compared to patients not receiving SDD (1-year mortality: 26.8% vs 20.4%, p = 0.056). Overall,

a difference in mortality is difficult to compare due to low incidences of 30-day mortality in all included studies. Hospital length of stay and ICU length of stay varied considerably regardless of SDD use. A clear benefit could not be demonstrated; in fact, some studies showed longer ICU and hospital stays in the group that received SDD. A mechanism explaining a longer ICU length of stay for patients receiving SDD warrants further investigation to understand the underlying factors and whether other patient-related or procedural variables may be contributing to this outcome."

Outcome assessment in esophagectomy suffers from a lack of standardization. In a systematic review examining 122 studies, 60.6% of papers did not define any of the measured complications [45]. Most of our included studies predate the publication of the International Consensus published by the Esophagectomy Complications Consensus Group (ECCG). This consensus group has published a standardized list of complications to be recorded after esophagectomy along with several uniform definitions to be used [46]. The study of Janssen et al., which is the most recent study in our analysis, was the only study in which these recommendations were adhered to. The studies included in our review lacked a uniform definition of pneumonia, anastomotic leakage and timing of mortality. The incidences of pneumonia and anastomotic leakage were pooled indiscriminately, representing the differences in real clinical practice [47].

The composition of different antibiotics used as components of SDD was not uniform in our included studies. We are therefore unable to make a statement about the ideal composition of SDD. The esophageal microbiome, which is believed to play a significant role in the development of postoperative complications, is influenced by the oral flora and has associations with various diseases, including esophageal cancer [48,49]. While previous studies have highlighted the important role of the microbiome in colon surgery, and research in pancreatic surgery has shown that the microbiome is altered by SDD, the effect of SDD on the diversity of the esophageal microbiome remains unclear [50,51]. Further investigation into how SDD impacts microbial diversity in the esophagus is warranted.

SDD has a few contraindications to its use, including a known allergy, sensitivity, or interaction with any of its components [29,52]. Antibiotic resistance may be a concern with the use of SDD; however, there are studies that have reported instances where such resistance has not been detected. For example, a randomized clinical trial evaluating the use of SDD for critically ill patients receiving mechanical ventilation showed a statistically significant reduction in antibiotic-resistant organisms in the SDD group without a difference in the incidence of new *C. difficile* infections [52]. Parenteral administration of cephalosporins and topical paste have not been associated with increased antibiotic resistance [52–55]. There are negligible systemic side effects as the topical paste reaches virtually no clinically significant bloodstream concentration in patients not undergoing renal replacement therapy [56].

## Conclusion

Although the data indicates that using SDD in patients undergoing an esophagectomy was associated with a lower incidence of postoperative pneumonia and anastomotic leakage, the available studies were not of sufficient quality to make a recommendation, given their age and risk of bias. A high-quality randomized controlled trial using standardized outcome definitions is needed to substantiate claims about SDD use in esophagectomy.

## Supporting information

**S1 File. PRISMA checklist.**
(DOCX)

**S2 File. Search Strategy.**
(DOCX)

**S3 File. Funnel plots.**
(DOCX)

**S4 File.  Supplementary tables, bias assessment.** ROBINS-I and RoB 2 assessment of included articles.
(XLSX)

## Acknowledgments

The authors would like to thank the librarians of the Erasmus Medical Center for their expertise and assistance with creating a search strategy and executing the literature search.

## Author contributions

**Conceptualization:** Marcus Paulus Buise.

**Data curation:** Sander Du X Oei, Jasper Gerrit  Jan Verbruggen.

**Formal analysis:** Sander Du X Oei, Jasper Gerrit  Jan Verbruggen, Sanne Elisabeth Hoeks.

**Funding acquisition:** Marcus Paulus Buise.

**Investigation:** Sander Du X Oei, Jasper Gerrit  Jan Verbruggen.

**Methodology:** Sander Du X Oei, Jasper Gerrit  Jan Verbruggen, Sanne Elisabeth Hoeks.

**Supervision:** Sanne Elisabeth Hoeks, Marcus Paulus Buise.

**Writing – original draft:** Sander Du X Oei, Jasper Gerrit  Jan Verbruggen.

**Writing – review & editing:** Sander Du X Oei, Jasper Gerrit  Jan Verbruggen, Sanne Elisabeth Hoeks, Marcus Paulus Buise.

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
