## [Decision Letter · Decision Letter 0]

PONE-D-24-39148Selective decontamination of the digestive tract in esophagectomy and the incidence of pneumonia and anastomotic leakage. A systematic review and meta-analysisPLOS ONE

Dear Dr. Oei,

Thank you for submitting your manuscript to PLOS ONE. After careful consideration, we feel that it has merit but does not fully meet PLOS ONE’s publication criteria as it currently stands. Therefore, we invite you to submit a revised version of the manuscript that addresses the points raised during the review process.

The reviewers acknowledge the well-executed review on using selective gut decontamination (SDD) to mitigate post-esophagectomy complications, despite the suboptimal quality of input studies. They recommend enhancing the justification for a randomized controlled trial (RCT), detailing SDD protocols, and considering the impact of evolving surgical techniques on outcomes. Additionally, the reviewers suggest addressing antibiotic practices, microbial diversity changes, and potential adverse events for a more comprehensive analysis.

We look forward to receiving your revised manuscript.

Kind regards,

Zubing Mei, MD,Ph.D

Academic Editor

PLOS ONE

Journal Requirements:

“M. Buise is part of an international think tank regarding minimally invasive esophagectomy which is supported by Medtronic. The other authors declare that they have no known conflicts of interest.”

3.  Please include captions for your Supporting Information files at the end of your manuscript, and update any in-text citations to match accordingly. Please see our Supporting Information guidelines for more information: http://journals.plos.org/plosone/s/supporting-information .

Reviewers' comments:

Reviewer's Responses to Questions

**Comments to the Author**

1. Is the manuscript technically sound, and do the data support the conclusions?

Reviewer #1: Yes

Reviewer #2: Yes

2. Has the statistical analysis been performed appropriately and rigorously? 

Reviewer #1: Yes

Reviewer #2: Yes

3. Have the authors made all data underlying the findings in their manuscript fully available?

Reviewer #1: Yes

Reviewer #2: Yes

4. Is the manuscript presented in an intelligible fashion and written in standard English?

Reviewer #1: Yes

Reviewer #2: Yes

5. Review Comments to the Author

Reviewer #1: Oei and colleagues from the Netherlands present a systematic review regarding the use of selective gut decontamination for reducing pneumonia and leak after oeospahgectomy. Over the years it has been challenging to gain traction in these areas with non-operative methods, mainly because of the surgical technique has been evolving, and the highly multifactorial nature of post-oesophagectomy pneumonia / acute lung injury and leaks. Overall it seems a well-executed review and thoughtfully written. The quality of the input studies are not great but I agree some insight comes from them. I couldn’t find much typographically error. It seems the goal is to develop a randomised study targeting the microbiome to reduce risk, which I would probably support given where we are now surgically (and collaboratively)

I’ve got a few comments for the authors that when addressed would strengthen the paper;

1. The key challenge you have in justifying an RCT is that reducing these risks seems to have no bearing at all on ITU stay or LOS. In some studies both were slightly longer in SDD.

2. In the introduction please provide some description regarding SDD. Is it oral or IV? Is it one dose or a week? How does it differ to the standard ‘on-induction’ IV doses. Understood there is a partial description in table 1

3. Every centre I have worked in provides 24-48h of antibiotic coverage for the operation and immediate post-op. How does the proposed strategies differ, and how has the IV prescription been considered in the presented studies (your description of the controls could be improved)

4. Because the antibiotics regimes are not well described, it is difficult to judge the risk of adverse events like diarrhoea, c. diff, carbapenamase producing infections etc. These aren’t reported. I disagree with the line that there is very little ill-effect from SDD use.

5. The discussion could be improved by checking the oesophageal microbiome literature for how these decontamination strategies actually change diversity

6. It would be worth pointing out that with advancing surgical technique (minimally invasive surgery, sparing of the pulmonary vagal branches, two lung ventilation, ERAS, prehabilitation) many of the precipitants of PPC have been considerably decreased, so potentially aspiration may have a proportionally bigger effect on acute lung injuries than when the first of the included studies were performed.

7. Following on from the previous point, at our centre we still preformed contrast studies to check for aspiration during swallowing, and delayed gastric conduit emptying. Our pneumonia rate is low. Thus if aspiration is considered to be the pathological mechanism, it would be good to understand why it is happening and treat the cause, rather than just treat the effects. This would need to be accounted for in a subsequent randomised studies

Minor:

1. Page 3 last four lines “Bacteria may interfere…” these lines are not well balanced. Undoubtedly there will be helpful bacteria also

2. Page 12 line 14 -16 non-significant lower risk please revise this to “the risk of anastomatic leakage in randomised trials was not significantly different”.

Reviewer #2: It is currently unknown whether selective decontamination of the digestive tract (SDD) affects incidence of postoperative pneumonia and anastomotic leakage in patients undergoing esophagectomy. The authors summarized current evidence regarding SDD in patients undergoing esophagectomy. Through searching various databases, they found that although the data indicates that using SDD in patients undergoing an esophagectomy was associated with a lower incidence of postoperative pneumonia and anastomotic leakage, the available studies were not of sufficient quality to make a recommendation, given their age and risk of bias. A high-quality randomized controlled trial using standardized outcome definitions is needed to substantiate claims about SDD use in esophagectomy.

Overall, the manuscript is well-written and easy follow. A few concerns need to be addressed before publishing.

1. Did the authors include Scopus database?

2. The selection criteria should be presented in details.

3. Strength and limitations should be included in the Discussion section.

6. PLOS authors have the option to publish the peer review history of their article (what does this mean? ). If published, this will include your full peer review and any attached files.

**Do you want your identity to be public for this peer review?** For information about this choice, including consent withdrawal, please see our Privacy Policy .

Reviewer #1: **Yes: ** Stefan Antonowicz

Reviewer #2: No

---

## [Author Response · Author response to Decision Letter 1]

27 Feb 2025

We sincerely appreciate the time and effort that the reviewers have dedicated to evaluating our work. We have carefully considered their comments and have made several revisions to the manuscript to address their concerns. We have provided a detailed point-by-point response to each of the reviewers’ comments. We hope that our adjustments can resolve any ambiguities and alleviate concerns regarding our research.

---

## [Decision Letter · Decision Letter 1]

May 23 2025

PONE-D-24-39148R1Selective decontamination of the digestive tract in esophagectomy and the incidence of pneumonia and anastomotic leakage. A systematic review and meta-analysisPLOS ONE

Dear Dr. Oei,

Thank you for submitting your manuscript to PLOS ONE. After careful consideration, we feel that it has merit but does not fully meet PLOS ONE’s publication criteria as it currently stands. Therefore, we invite you to submit a revised version of the manuscript that addresses the points raised during the review process.

Please response to the minor comments raised by the reviewer regarding some manuscript details.

We look forward to receiving your revised manuscript.

Kind regards,

Zubing Mei, MD,PH.D

Academic Editor

PLOS ONE

Journal Requirements:

Reviewers' comments:

Reviewer's Responses to Questions

**Comments to the Author**

1. If the authors have adequately addressed your comments raised in a previous round of review and you feel that this manuscript is now acceptable for publication, you may indicate that here to bypass the “Comments to the Author” section, enter your conflict of interest statement in the “Confidential to Editor” section, and submit your "Accept" recommendation.

Reviewer #1: All comments have been addressed

2. Is the manuscript technically sound, and do the data support the conclusions?

Reviewer #1: Yes

3. Has the statistical analysis been performed appropriately and rigorously? 

Reviewer #1: Yes

4. Have the authors made all data underlying the findings in their manuscript fully available?

Reviewer #1: Yes

5. Is the manuscript presented in an intelligible fashion and written in standard English?

Reviewer #1: Yes

6. Review Comments to the Author

Reviewer #1: Thanks for your work

Please double-check the figure / table legends before finalising

Also please delete this sentence from the discussion which i think is non-contributory: "Among the strategies mentioned, minimally invasive surgery, ERAS, and prehabilitation are wellestablished in routine clinical practice. The anticipated beneficial effects of vagal-sparing surgical

techniques are attributed to the vagus nerve's role in modulating anti-inflammatory responses [40, 41].

Formatted: Heading 1, Line spacing: single

14

However, this technique remains contentious, and its application is currently limited to a small cohort

of patients with early-stage cancers or severe epithelial dysplasia."

7. PLOS authors have the option to publish the peer review history of their article (what does this mean? ). If published, this will include your full peer review and any attached files.

**Do you want your identity to be public for this peer review?** For information about this choice, including consent withdrawal, please see our Privacy Policy .

Reviewer #1: No

---

## [Author Response · Author response to Decision Letter 2]

20 Apr 2025

Our response to reviewers document has been uploaded.

In short, we have checked the references and that figures were submitted through PACE, we have double-checked figures and table legends and made adjustments to better align with the PLOS ONE submission guidelines and we have removed a sentence from the discussion that was non-contributory.

We think that the revisions we made in response to the reviewers’ comments have improved the manuscript. We would like to express our sincere gratitude for the valuable feedback, and we hope that the revised manuscript will now meet the standards of PLOS ONE.

We are looking forward to your consideration of our revised submission and are happy to provide any further clarifications if needed.

Sincerely,

Sander Oei

On behalf of all authors.

---

## [Editor Report · Decision Letter 2]

Selective decontamination of the digestive tract in esophagectomy and the incidence of pneumonia and anastomotic leakage. A systematic review and meta-analysis

PONE-D-24-39148R2

Dear Dr. Oei,

We’re pleased to inform you that your manuscript has been judged scientifically suitable for publication and will be formally accepted for publication once it meets all outstanding technical requirements.

Kind regards,

Zubing Mei, MD,Ph.D

Academic Editor

PLOS ONE
---

## [Editor Report · Acceptance letter]

PONE-D-24-39148R2

PLOS ONE

Dear Dr. Buise,

I'm pleased to inform you that your manuscript has been deemed suitable for publication in PLOS ONE. Congratulations! Your manuscript is now being handed over to our production team.

Kind regards,

on behalf of

Dr. Zubing Mei

Academic Editor

PLOS ONE